# Biomarkers of Coagulation and Inflammation in Dogs after Randomized Administration of 6% Hydroxyethyl Starch 130/0.4 or Hartmann’s Solution

**DOI:** 10.3390/ani12192691

**Published:** 2022-10-06

**Authors:** Corrin J. Boyd, Anthea L. Raisis, Claire R. Sharp, Melissa A. Claus, Giselle Hosgood, Lisa Smart

**Affiliations:** 1School of Veterinary Medicine, Murdoch University, Perth, WA 6150, Australia; 2Harry Butler Institute, Murdoch University, Perth, WA 6150, Australia; 3Perth Veterinary Specialists, Perth, WA 6017, Australia; 4Small Animal Specialist Hospital, Tuggerah, NSW 2259, Australia

**Keywords:** activated partial thromboplastin time, coagulation factor, colloid, crystalloid, cytokine, fibrinogen, fluid therapy, hemostasis, prothrombin time, von Willebrand factor

## Abstract

**Simple Summary:**

Critically ill dogs often require intravenous fluid therapy to improve circulation and treat life-threatening shock. Standardly, a balanced electrolyte solution called a crystalloid is prescribed. Some veterinarians also prescribe colloid fluids like hydroxyethyl starch (HES), which are crystalloids that contain large starch molecules, as more of the volume delivered stays in circulation compared to crystalloid. However, there is concern that colloids may impair blood clot formation and increase the risk of serious bleeding. To better understand this risk, this study compared the function of individual clotting components within blood samples collected over time from critically ill dogs that received either crystalloids or HES. The blood samples were saved from a previous clinical trial where critically ill dogs requiring intravenous fluids were randomized to receive HES or crystalloids. Blood was collected before fluid therapy and then 6, 12, and 24 h after the start of the intervention fluid. The results did not provide evidence that HES impaired clotting more than crystalloids. However, considering the small sample size and variability between dogs, this result does not provide evidence that HES is safe. Larger or more targeted studies are required to further assess the effect of HES on blood clotting in dogs.

**Abstract:**

Synthetic colloid fluids containing hydroxyethyl starch (HES) have been associated with impairment of coagulation in dogs. It is unknown if HES causes coagulation impairment in dogs with naturally occurring critical illness. This study used banked plasma samples from a blinded, randomized clinical trial comparing HES and balanced isotonic crystalloid for bolus fluid therapy in 39 critically ill dogs. Blood was collected prior to fluid administration and 6, 12, and 24 h thereafter. Coagulation biomarkers measured at each time point included prothrombin time, activated partial thromboplastin time, thrombin time, fibrinogen concentration, and the activities of coagulation factors V, VII, VIII, IX, and X, von Willebrand factor antigen, antithrombin, and protein C. Given the links between coagulation and inflammation, cytokine concentrations were also measured, including interleukins 6, 8, 10, and 18, keratinocyte-derived chemokine, and monocyte chemoattractant protein-1. Data were analyzed with linear mixed effects models. No significant treatment-by-time interactions were found for any biomarker, indicating that the pattern of change over time was not modified by treatment. Examining the main effect of time showed significant changes in several coagulation biomarkers and keratinocyte-derived chemokines. This study could not detect evidence of coagulation impairment with HES.

## 1. Introduction

Isotonic crystalloid fluids and hydroxyethyl starch (HES)-containing colloid solutions are both currently used for intravenous bolus fluid therapy in dogs [1,2]. Although HES fluids may be a more efficient volume expander than isotonic crystalloid [3] and have the theoretical benefit of mitigating fluid extravasation, HES administration in critically ill people is not associated with improved survival [4]. Furthermore, the use of HES has been associated with increased bleeding risk in some human-patient populations, including with the use of low molecular weight (MW) HES fluids, which are claimed to have a lower risk of coagulation impairment compared to high MW HES products [4,5,6].

Synthetic colloid solutions may adversely affect coagulation by interfering with platelet function, reducing coagulation factor concentrations, decreasing clot strength, and increasing clot friability [7]. Evidence of coagulation impairment beyond the expected effects of hemodilution has been demonstrated for low MW HES in experimental studies in dogs, for both in vitro [8,9] and in vivo models of hemorrhage [10] and sepsis [11]. Clinical evidence is limited to a single clinical trial in 42 dogs with spontaneous hemoperitoneum, with greater hypocoagulability demonstrated in dogs immediately after receiving low MW HES compared to those receiving a balanced isotonic crystalloid [12]. Interestingly, differences between groups were only detected in viscoelastic testing. There were no differences detected in plasma coagulation times and fibrinogen concentration, but a detailed analysis of plasma-based coagulation was not performed. It is unknown if these findings can be extrapolated to a heterogenous cohort of critically ill dogs, if these effects change over time, or if there are effects on individual plasma coagulation factors such as factor VIII or von Willebrand factor [13].

Coagulation and inflammation have complex interactions in the critically ill patient. Therefore, any treatment effect of crystalloid or HES fluid on coagulation may be altered by baseline inflammation. Furthermore, different fluids may promote or dampen inflammatory responses. For example, HES mitigated acute inflammatory responses in rodent models of sepsis and hemorrhage [14,15]. In one canine experimental study, large-volume crystalloid administration for hemorrhagic shock was pro-inflammatory, while low MW HES administration was not [16]. To the authors’ knowledge, no clinical studies in dogs have evaluated the effects of fluid type on inflammation.

This study evaluated biomarkers of coagulation and inflammation in banked plasma samples from a blinded, randomized clinical trial that compared urine biomarkers of acute kidney injury in dogs prescribed a fluid bolus for naturally occurring clinical conditions [17]. The objective was to compare changes over time in the plasma biomarkers of coagulation and inflammation after bolus administration of 6% HES 130/0.4 or a crystalloid fluid. We hypothesized that plasma from dogs treated with HES would have greater hypocoagulability and lower concentrations of inflammation biomarkers compared to plasma from dogs treated with crystalloid. Furthermore, we hypothesized that hypocoagulability and increases in inflammation biomarkers would occur over time in all samples.

## 2. Materials & Methods

### 2.1. Study Design

This study utilized banked plasma samples from the NGAL and Cystatin C After Hartmann’s or Starch (NACHOS) clinical trial; methods previously reported [17]. Briefly, NACHOS was a single-center, prospective, blinded, randomized, parallel-group, preliminary clinical trial. The study was approved by the institutional Animal Ethics Committee (R2964/17) and was conducted in accordance with the Australian Code for the Care and Use of Animals for Scientific Purposes. Informed client consent was obtained prior to enrollment. The reporting of this trial adhered to the CONSORT guidelines, [18] with the required patient flow diagram and baseline data being presented in the original publication [17].

Dogs under the care of the emergency and critical care department were eligible for inclusion if they were prescribed a fluid bolus of at least 10 mL/kg over less than 30 min for any clinical condition at any point during hospitalization. Several exclusion criteria were applied to enroll an appropriate cohort for the study of kidney injury: age < 6 months, administration of a synthetic colloid or nephrotoxic medication within the previous 2 weeks, known pre-existing kidney disease or pyuria, hemorrhagic shock requiring immediate blood product administration, moderate or severe interstitial dehydration, expectation of imminent death or euthanasia, expectation of hospital stay less than 24 h, or previous enrollment in the study [17].

Enrolled dogs were randomized to receive either 6% HES 130/0.4 in a balanced crystalloid carrier (Volulyte, Fresenius Kabi, Bad Homburg, Germany) or the balanced isotonic crystalloid Hartmann’s solution (Compound Sodium Lactate, Baxter Healthcare, Deerfield, IL, USA) (CRYST). Computer-generated randomized group allocations were stored in sequentially numbered sealed envelopes. Treating veterinarians, nurses, and investigators were blinded to treatment by use of an opaque bag covering the study fluid bag. An initial bolus of 10 mL/kg of randomized study fluid was administered over less than 30 min. The study fluid was used for further fluid boluses within the subsequent 24 h period up to a maximum of 40 mL/kg total volume. Other crystalloid fluids or blood products were allowed at any time for purposes other than blood volume expansion.

### 2.2. Sample Collection

Baseline demographic data were collected as previously described [17]. Additionally, medical records were reviewed for data on surgery or any blood product transfusion during hospitalization either before or after study enrollment. Administration of antiplatelet or anticoagulant medications prior to or during the 24 h sampling period was recorded. Baseline serum C-reactive protein concentration, measured for a concurrent study that utilized these samples as a comparison group, [19] was recorded when available. Blood samples were collected prior to the initial study fluid bolus (baseline) and at 6 (T6), 12 (T12), and 24 (T24) hours thereafter. Blood was collected using a three-syringe technique from an indwelling venous or arterial cannula, if possible. Otherwise, venepuncture of the jugular vein or a peripheral vein was performed. Blood was immediately transferred into EDTA and 3.2% sodium citrate tubes. Citrate tubes were filled with an anticoagulant:blood ratio of 1:9, resulting in a final citrate concentration of 10.8 mmol/L. Samples were stored on ice packs for up to 60 min, then were centrifuged at 1350× *g* for 10 min. The plasma was separated into aliquots and stored at −80 °C until analysis.

### 2.3. Coagulation Biomarker Measurement

Prothrombin time (PT), activated partial thromboplastin time (APTT), thrombin time (TT), fibrinogen concentration, and activities of coagulation factors V (FV), VII (FVII), VIII (FVIII), IX (FIX), X (FX), von Willebrand factor antigen (vWF), antithrombin (AT), and protein C (PC) were measured in citrated plasma using a commercial turbidimetric analyzer (ACL-TOP 300, Instrumentation Laboratory, Werfen, Artarmon, Australia). The coagulation times, PT, APTT, and TT were measured using the manufacturer’s reagents. Fibrinogen concentration was measured using the manufacturer’s reagent for the Clauss method, calibrated using the manufacturer’s calibration plasma. All remaining coagulation factor activities were calibrated with canine pooled plasma, as previously described [20]. Activity of FV, FVII, and FX were measured using a modified PT test, while FVIII and FIX were measured using a modified APTT test, all using the manufacturer’s specific factor-depleted plasmas. A latex-enhanced immunoassay was used to measure vWF. Automated chromogenic assays were used to measure AT and PC. Samples were thawed in batches using a temperature-controlled water bath at 37 °C. Assay order was determined by the analyzer’s high-throughput algorithm. Quality controls were run daily. Coagulation results were compared to published reference intervals where available [21].

### 2.4. Inflammation Biomarker Measurement

Interleukin 6 (IL6), interleukin 8 (IL8), interleukin 10 (IL10), interleukin 18 (IL18), keratinocyte-derived chemokine (KC), and monocyte chemoattractant protein-1 (MCP) (also known as chemokine (C-C motif) ligand 2 (CCL2)) were measured in EDTA plasma using a commercial magnetic bead multiplexed assay kit (MILLIPLEX MAP Canine Cytokine Magnetic Bead Panel, MilliporeSigma, Burlington, MA, USA). Biomarkers were selected based on a previous study in canine hemorrhagic shock [16] and for consistency with a concurrent study on canine anaphylaxis [19]. The assay was performed according to the manufacturer’s instructions, with a sample dilution of 1:2. Samples were assayed in duplicate and repeated if the coefficient of variation was greater than 20%. Samples above the upper limit of detection of the assay were repeated at a 1:10 dilution. Samples below the lower limit of detection of the assay, but with fluorescence above blank, were repeated undiluted. Samples where the final measurement was below the lower limit of detection were assigned the value:

Lower limit of detection/$\sqrt{2}$.

### 2.5. Sample Size Calculation

This study was considered exploratory and sample size was dictated by the availability of samples from a previous clinical trial. However, a power calculation based on data from a canine hemorrhagic shock study [10] determined that a sample size of 16 in each arm will give 80% power (α = 0.05) to detect a 10% difference in vWF activity between groups with an estimated standard deviation of 10%.

### 2.6. Statistical Methods

Statistical analysis was performed on fully unblinded data using commercially available statistical software (SAS Version 9.4, SAS Institute, Cary, NC, USA). Continuous numeric baseline characteristics were assessed for normality by visualization of histograms and Q-Q plots and are summarized as mean ± standard deviation, ordinal baseline characteristics are summarized as median (minimum–maximum), and categorical baseline characteristics are summarized as counts. Biomarkers were analyzed using linear mixed effects models with the fixed effects of group, time, and their interaction and a random variance of dog nested within treatment. The residuals were assessed for normality as above, and right-skewed data were log transformed to approximate a normal distribution. Data were summarized as mean (95% confidence interval) for normally distributed data or geometric mean (95% confidence interval) for right-skewed data. Additional non-parametric summary statistics are provided as a supplement. Additional analysis was performed with the inclusion of potential covariates: reason for admission, presence of sepsis, Acute Patient Physiologic and Laboratory Evaluation fast (APPLE_fast_) illness severity score, [22] and volume of study fluid administered. Significance was set at *p* < 0.05.

## 3. Results

A total of 40 dogs were included in the clinical trial, with 21 randomized to HES and 19 randomized to CRYST. The participant flow chart and baseline characteristics of this cohort have been previously described in detail [17]. One dog from the CRYST group was excluded from all biomarker measurements due to the unavailability of plasma samples (Appendix A). The baseline characteristics of the remaining 39 dogs are briefly summarized below. The mean age was 6.1 ± 3.7 years for HES and 6.8 ± 4.3 years for CRYST. Illness severity was moderate, with median (minimum–maximum) APPLE_fast_ scores of 24 (14–34) for HES and 23.5 (19–38) for CRYST. The dogs were treated for a range of diseases, with gastrointestinal disease being the most frequent (11 HES, 13 CRYST). Some dogs had conditions considered a high risk for coagulopathy and hemorrhage: sepsis (5 HES, 2 CRYST), anaphylaxis (2 HES, 1 CRYST), trauma (1 HES, 2 CRYST), hemoperitoneum due to neoplasia (1 HES, 1 CRYST), and acute hepatic injury from cycad toxicosis (1 HES). Surgery was performed during hospitalization in five dogs for HES and eight dogs for CRYST, mostly celiotomy. No dogs were administered antiplatelet or anticoagulant medications prior to or during sample collection. The mean volume of study fluid administered over the 24 h following enrollment was 23.1 ± 10.0 mL/kg for HES and 25.1 ± 14.6 mL/kg for CRYST. Baseline serum C-reactive protein concentration (*n* = 27) was above the reference upper limit of 10 mg/L in 14/15 dogs for HES (mean 107.4 ± 91.2 mg/L) and 9/12 dogs for CRYST (mean 84.7 ± 90.7 mg/L). The blood transfusion data are summarized in Table 1. All seven dogs that were administered transfusions received all blood products after study enrollment, with six dogs being administered blood products during the first 24 h after enrollment. No dogs were administered human or canine albumin solution.

The coagulation and inflammation biomarker results are shown in Figure 1 and Figure 2, respectively, and are numerically summarized in Appendix A. Some samples were not collected after T6 due to death or discharge from hospital, and some samples were of insufficient volume for a complete panel of biomarkers (Appendix A). Log transformation was performed for analysis of all inflammation biomarkers (IL6, IL8, IL10, IL18, KC, and MCP). There were no significant treatment-by-time interaction effects, indicating that the pattern of change over time was not modulated by treatment for any biomarker (Table 2). There was a significant main effect of time, indicating a change over time across all samples for PT, fibrinogen concentration, FV, FVII, FVIII, FIX, FX, vWF, AT, PC, and KC (Table 2). The pattern of change was for an increase in coagulation times and a decrease in coagulation factor activities over time. There was a significant main effect of treatment for APTT; this indicates a difference between groups regardless of time and this was driven by the difference between groups at baseline (Table 2). The models with the additional variables of reason for admission, presence of sepsis, APPLE_fast_ score, and volume of study fluid administered showed similar results (Appendix A).

## 4. Discussion

Plasma from dogs randomized to receive HES or CRYST for bolus fluid therapy showed no differences in terms of how coagulation factor activities and inflammation biomarker concentrations changed over time. Although a relatively small sample size was examined, no covariates were identified that might have confounded these results. Regardless of treatment, there were significant changes over 24 h in several coagulation biomarkers and the inflammatory biomarker KC.

The failure to detect differences between HES and CRYST in modulating coagulation biomarkers over time was contrary to our hypothesis that proposed HES would be associated with greater hypocoagulability. This was unexpected for two reasons: HES causes a dilutional coagulopathy and HES has direct anticoagulant effects. Firstly, it was anticipated that HES would be associated with more hemodilution and a direct effect on hypocoagulability. With an equal dose, HES can cause more hemodilution than crystalloids for at least 4–6 h due to decreased extravasation, with a theoretical volume efficiency of 3:1 to 4:1 [3,23]. Therefore, in our study, given equal volume administrations (HES 23.5 ± 5.9 mL/kg; CRYST 23.5 ± 4.4 mL/kg), more hemodilution would be expected in the HES group. However, this may not be the case. Similar total doses of crystalloid and HES have been used in human randomized controlled trials with patients resuscitated to clinical endpoints, supporting a hypothesis that the increased volume efficiency of HES may not be as relevant in critical illness due to glycocalyx shedding and increased vascular permeability [24,25,26]. Furthermore, in our original study, other non-bolus fluid therapy, such as for rehydration, may have contributed to hemodilution in either group. Since markers of hemodilution were not measured, this could not be quantified.

The second premise for our hypothesis of more hypocoagulability with HES was based on other, non-dilutional, mechanisms by which HES can cause hypocoagulability [7]. These include platelet dysfunction, [27] deficiency of vWF and FVIII, [13] generalized coagulation factor deficiencies, [28] altered fibrin meshwork structure [29] with impaired crosslinking, [30] and enhanced fibrinolysis [31]. The selection of coagulation assays in our study was limited to those suitable for frozen samples, which prevented the analysis of some mechanisms of hypocoagulability. Effects on platelet function would not have been detected in our study, as only the plasma coagulation components were assessed. Likewise, detecting effects on fibrin meshwork strength and fibrinolysis would have required other testing modalities such as viscoelastic assays. Such effects were detected in a clinical trial in dogs with spontaneous hemoperitoneum, where dogs treated with HES had lower maximum clot firmness in the EXTEM assay of rotational thromboelastometry compared to dogs treated with crystalloid [12]. This was despite minimal changes in plasma-based assays. Therefore, there may have been important undetected differences in our study due to limitations on assay selection. We were unable to detect any differences in coagulation factor activities, especially for FVIII or vWF, despite these being known to be affected by HES [13]. This may be due to an insufficient dose of HES, lower susceptibility of dogs to these effects compared to other species, or a decreased risk with low MW HES compared to other HES products. Alternatively, it may represent type 2 error given the large variability in coagulation factor activity between the subjects in our cohort. Baseline variability was noted within treatment groups, likely a result of the wide range and severity of underlying conditions. Accounting for the contribution of variance inherent to the subject (dog) nested within the treatment groups by using the mixed model may have addressed some of this heterogeneity. Further exploration did not identify any covariates that might have confounded our results. Thus, although no evidence of a treatment effect was detected under the conditions of our study, a larger sample size, with stratified randomization across possible covariates such as sepsis [11,32] or trauma [33,34], would be necessary to verify these results.

The failure to detect differences between HES and CRYST in the pattern of change of inflammatory biomarker concentration was also contrary to our hypothesis. Experimental studies suggest an anti-inflammatory effect of HES due to down-regulation of the nuclear factor kappa B pathway [14,15]. These studies were performed in tightly controlled models of hemorrhage and sepsis and may not translate to clinically relevant differences in naturally occurring disease. This is supported by human clinical studies in major surgery [35] and sepsis [36], which did not detect any differences in inflammatory biomarkers when HES was compared to crystalloid. A canine study investigating cytokine concentrations following bolus fluid therapy for experimental hemorrhagic shock found higher plasma cytokine concentrations with crystalloid administration compared with HES [16]. However, unlike in our study, where the dose of fluid in both groups was similar, the total dose of crystalloid administered in the experimental study was four-fold higher than the dose of HES (80 mL/kg compared with 20 mL/kg). Therefore, it is not clear if the difference in plasma cytokine concentrations was due to the anti-inflammatory properties of HES or if the much larger volume of crystalloid administered was pro-inflammatory. Finally, both the crystalloid and HES fluids were acetate-buffered in the experimental study above [16] compared to the use of acetate-buffered HES and lactate-buffered CRYST in our study. Acetate may be pro-inflammatory [37], which could have offset any anti-inflammatory effect of HES in our study.

This is the first study to document changes in coagulation for 24 h after bolus intravenous fluid therapy in critically ill dogs. Significant changes over time were detected, with increasing PT and decreasing coagulation factor activities suggesting a pattern towards hypocoagulability in both groups. Both HES and crystalloid fluids can cause a dilutional coagulopathy proportionate to the degree of hemodilution [7,8]. It is likely that hemodilution increased over time in our study due to the administration of bolus study fluid and other fluid therapy. Unfortunately, this cannot be confirmed or quantified as we did not measure a specific marker of hemodilution, such as hemoglobin concentration. A potential alternate reason for the trend towards hypocoagulability in both groups is coagulation factor consumption, loss, or decreased production due to the underlying critical illness. Hypocoagulability due to several mechanisms has been documented in conditions such as sepsis [32], trauma [33,34], hemoperitoneum [38], anaphylaxis [39], and acute hepatic failure [40], all of which were represented in dogs within our cohort. Regardless of the underlying cause, the pattern seen in our study supports the assertion that critically ill dogs may exhibit hemorrhage in the first 24 h after bolus fluid therapy. Progressive hypocoagulability is associated with non-survival in human critical illness [41]. Our sample size was too small and the duration of observation was too short to meaningfully assess whether hypocoagulability was associated with non-survival.

We found no evidence of change in inflammation biomarkers, except KC, which showed a significant decrease over time. This is contrary to our hypothesis. Increases in circulating cytokine concentrations have been documented in the first 3 h following fluid administration in experimental canine hemorrhagic shock [16]. Thus, a peak in cytokine concentrations may have been missed in our study, as the first time point after bolus initiation was at 6 h. Cytokine concentrations following fluid resuscitation and major surgery in humans peak immediately postoperatively and decrease in subsequent hours to days [35,42]. Cytokine concentrations also decrease from baseline at 24–48 h after fluid resuscitation for sepsis in humans [36].

The strengths of our study include the blinded, randomized design, the inclusion of a cohort of dogs with conditions where there is equipoise regarding HES fluid administration, the use of a panel of multiple biomarkers, and the relatively high volume of study fluid administered, as HES-associated hypocoagulability in dogs may be dose-dependent [8,43]. The fluid dose was chosen for the original kidney injury study [17] but was considered to be appropriate for the investigation of coagulation. However, there were also several limitations, and as such, the study should be considered exploratory. The sample size was dictated by a previous clinical trial. Given the heterogeneity of the cohort, there was substantial variability in several biomarkers, which may have led to type 2 error. As the study used stored plasma samples, the assessment of the platelet contribution to coagulation was not possible. The impairment of platelet function by HES has been documented in dogs [43]. The assessment of platelet function would require specific platelet function assays, such as the platelet function analyzer-100 closure time or aggregometry or global assays of coagulation such as thromboelastography or rotational thromboelastometry. Global assays also would have allowed for the detection of alterations in fibrin meshwork strength and fibrinolysis. These assays require fresh citrated whole blood, and therefore must be performed immediately after sample collection, which was not possible in our study. We did not measure any markers of hemodilution, such as hemoglobin concentration, and so we cannot determine whether significant changes over time were due to hemodilution or another mechanism. Data on transfusion administration were not statistically analyzed for differences between groups due to the insufficient sample size. Furthermore, the administration of plasma transfusion to some dogs may have affected the subsequent coagulation test results. These dogs were not excluded, as doing so would introduce bias. The limited sampling time points may have missed differences between fluid types immediately after administration, which has been documented in other studies [10,12].

## 5. Conclusions

This study showed that dogs with naturally occurring conditions requiring bolus intravenous fluid therapy had significant changes in biomarkers of coagulation and inflammation over 24 h, primarily trending towards hypocoagulability. This study was unable to detect differences between HES and CRYST. However, given the small sample size and substantial variability between subjects, this finding should be verified in larger studies with the ability to stratify across covariates such as underlying conditions. Future clinical trials should consider additional time points soon after fluid administration, the use of viscoelastic and platelet function tests, and the measurement of a marker of hemodilution.

## Figures and Tables

**Figure 1 animals-12-02691-f001:**
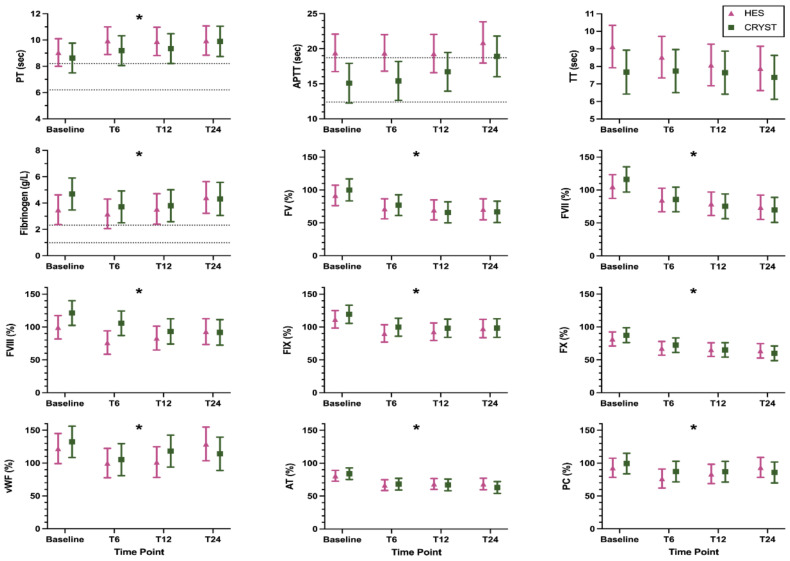
Coagulation biomarkers (mean, 95% confidence interval) in dogs randomized to receive 6% hydroxyethyl starch 130/0.4 (HES) or Hartmann’s solution (CRYST). Dotted lines indicate published reference intervals where available. * indicates a significant (*p* < 0.05) change over time where both groups are considered together. Abbreviations: APTT, activated partial thromboplastin time; AT, antithrombin activity; FV, factor V activity; FVII, factor VII activity; FVIII, factor VIII activity; FIX, factor IX activity; FX, factor X activity; PC, protein C activity; PT, prothrombin time; TT, thrombin time; vWF, von Willebrand factor antigen.

**Figure 2 animals-12-02691-f002:**
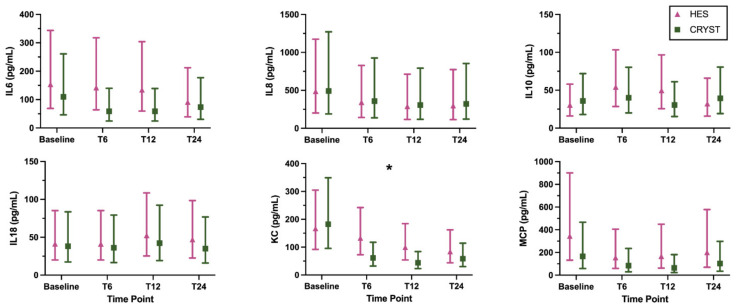
Inflammation biomarkers (geometric mean, 95% confidence interval) in dogs randomized to receive 6% hydroxyethyl starch 130/0.4 (HES) or Hartmann’s solution (CRYST). * indicates a significant (*p* < 0.05) change over time where both groups are considered together. Abbreviations: IL6, interleukin 6; IL8, interleukin 8; IL10, interleukin 10; IL18, interleukin 18; KC, keratinocyte-derived chemokine; MCP, monocyte chemoattractant protein-1.

**Table 1 animals-12-02691-t001:** Blood transfusions administered during hospitalization to dogs randomized to receive 6% hydroxyethyl starch 130/0.4 (HES) or Hartmann’s solution (CRYST). All transfusions were administered after study enrollment.

Transfusion Type	HES	CRYST
No transfusion (*n*)	16	16
Packed red blood cells only (*n*)	1	0
Plasma only (*n*)	2	0
Both packed red blood cells and plasma (*n*)	2 *	2 ^#^

* One of these dogs also received platelet-rich plasma. ^#^ One of these dogs also received an autotransfusion.

**Table 2 animals-12-02691-t002:** Fixed effect *p*-values for linear mixed effects models analyzing coagulation and inflammation biomarkers over time in dogs randomized to receive 6% hydroxyethyl starch 130/0.4 or Hartmann’s solution. These models contained a random variance of dog, nested within treatment, and no additional covariates. Significant results (*p* < 0.05) are in bold.

Biomarker	Time Main Effect	Treatment Main Effect	Treatment-by-Time Interaction Effect
Prothrombin time	0.002	0.53	0.68
Activated partial thromboplastin time	0.12	**0.025**	0.72
Thrombin time	0.39	0.24	0.68
Fibrinogen concentration	**0.029**	0.54	0.24
Factor V activity	**<0.001**	0.87	0.46
Factor VII activity	**<0.001**	0.94	0.36
Factor VIII activity	**0.006**	0.15	0.16
Factor IX activity	**<0.001**	0.49	0.76
Factor X activity	**<0.001**	0.81	0.45
von Willebrand factor antigen	**0.028**	0.74	0.39
Antithrombin activity	**<0.001**	0.92	0.42
Protein C activity	**<0.001**	0.75	0.08
Interleukin 6 *	0.09	0.31	0.22
Interleukin 8 *	0.27	0.93	1.00
Interleukin 10 *	0.47	0.78	0.37
Interleukin 18 *	0.17	0.73	0.69
Keratinocyte-derived chemokine *	**<0.001**	0.23	0.06
Monocyte chemoattractant protein-1 *	0.053	0.22	0.96

* Data were log transformed for analysis.

## Data Availability

Detailed summary data are provided in the manuscript. The raw data supporting the conclusions of this manuscript will be made available by the authors, without undue reservation, to any qualified researcher upon request.

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
