# Peer review of "Biomarkers of Coagulation and Inflammation in Dogs after Randomized Administration of 6% Hydroxyethyl Starch 130/0.4 or Hartmann’s Solution"

_animals, 2022, doi:10.3390/ani12192691_

Round 1
Reviewer 1 Report
this is a well written report of the effects of 2 fluids on coagulation and inflammation bio markers in a limit number of dogs. There is concern about Table 1 which shows that 2 dogs received plasma and 2 dogs whole blood transfusion in the HES group and 2 of the dogs received PRP and/or an autotransfusion; these would have contained the factors tested for. Perhaps there should be a comment in the discussion about these or they should be excluded. The list of references is very long and may of t he ones form 25 - 49 are not directly related to what was done in this study perhaps some of them can be deleted.
Author Response
The authors thank the reviewer for your kind comments in your review.
Thank you for noting the limitation of the effect of transfusion on coagulation results. We have added the following sentences to our discussion:
“Furthermore, the administration of plasma transfusion to some dogs may have affected the subsequent coagulation test results. These dogs were not excluded as doing so would introduce bias.”
We have reduced the number of references by citing only key papers for points where we previously cited multiple papers. This has reduced the overall reference count from 49 to 43.
Reviewer 2 Report
Dear Authors,
The submitted paper provides unique information with scientific interest in the field of canine coagulopathies and fluid administration and should be accepted for publication at the special issue of Animals. The paper is well written with appropriate figures and supplemental material. Although limitations exist mostly due to the size, disease severity, and design of the study (use of banked plasma samples), these are clearly stated and in length discussed. I have only minor revisions to propose.
Note to authors: Direct citation of the text is written between ellipses and in italic and any additions to original text in blue.
Line 29: Please add the final number of dogs included in the statistical analysis. …..in 39 critically ill dogs…
Line 38: Due to the limitations of the study, please use ..could not detect.. instead of ..did not..
Line 146: please add the common alternative name of MCP-1 … monocyte chemoattractant protein-1 (MCP) (also known as chemokine (C-C motif) ligand 2 (CCL2)).
Line 201: Please specify if human or canine albumin solution was administrated during the first 24h after enrolment.
Line 259: Please add a full stop after ...group. However,…
Line 332: Please add a full stop after …time. This is…
Author Response
The authors thank the reviewer for your kind comments in your review. We have made all your requested changes. No albumin solution was administered; this information has been included in the manuscript.